# A Continuous Markov-Chain Model for the Simulation of COVID-19 Epidemic Dynamics

**DOI:** 10.3390/biology11020190

**Published:** 2022-01-26

**Authors:** Zhaobin Xu, Hongmei Zhang, Zuyi Huang

**Affiliations:** 1Department of Life Science, Dezhou University, Dezhou 253023, China; dzxy7678@163.com; 2Department of Chemical and Biological Engineering, Villanova University, Villanova, PA 19085, USA

**Keywords:** Markov-chain model, COVID-19, reproduction number, mutation, herd immunity threshold

## Abstract

**Simple Summary:**

Predicting the spreading trend of the COVID-19 epidemic is one of the hot topics in the modeling field. In this study, we applied a continuous Markov-chain model to simulate the spread of the COVID-19 epidemic. The results of this study indicate that the herd immunity threshold should be significantly higher than 1 − 1/*R*_0_. Taking the immunity waning effect into consideration, the model could predict an epidemic resurgence after the herd immunity threshold. Meanwhile, this Markov-chain approach could also forecast the epidemic distribution and predict the epidemic hotspots at different times. It is implied from our model that it is significantly challenging to eradicate SARS-CoV-2 in the short term. The actual epidemic development is consistent with our prediction. In the end, this method displayed great potential as an alternative approach to traditional compartment models.

**Abstract:**

To address the urgent need to accurately predict the spreading trend of the COVID-19 epidemic, a continuous Markov-chain model was, for the first time, developed in this work to predict the spread of COVID-19 infection. A probability matrix of infection was first developed in this model based upon the contact frequency of individuals within the population, the individual’s characteristics, and other factors that can effectively reflect the epidemic’s temporal and spatial variation characteristics. The Markov-chain model was then extended to incorporate both the mutation effect of COVID-19 and the decaying effect of antibodies. The developed comprehensive Markov-chain model that integrates the aforementioned factors was finally tested by real data to predict the trend of the COVID-19 epidemic. The result shows that our model can effectively avoid the prediction dilemma that may exist with traditional ordinary differential equations model, such as the susceptible–infectious–recovered (SIR) model. Meanwhile, it can forecast the epidemic distribution and predict the epidemic hotspots geographically at different times. It is also demonstrated in our result that the influence of the population’s spatial and geographic distribution in a herd infection event is needed in the model for a better prediction of the epidemic trend. At the same time, our result indicates that no simple derivative relationship exists between the threshold of herd immunity and the virus basic reproduction number *R*_0_. The threshold of herd immunity achieved through natural immunity is significantly higher than 1 − 1/*R*_0_. These not only explain the theoretical misconceptions of herd immunity thresholds in herd immunity theory but also provide a guidance for predicting the optimal vaccination coverage. In addition, our model can predict the temporal and spatial distribution of infections in different epidemic waves. It is implied from our model that it is challenging to eradicate COVID-19 in the short term for a large population size and a wide spatial distribution. It is predicted that COVID-19 is likely to coexist with humans for a long time and that it will exhibit multipoint epidemic effects at a later stage. The statistical evidence is consistent with our prediction and strongly supports our modeling results.

## 1. Introduction

By 1 January 2022, the COVID-19 epidemic had caused 290 million infections and more than 5.4 million deaths worldwide. This makes it the most significant public health crisis faced by the world since World War II. It is of significant value to predict the trend of the COVID-19 epidemic situation. Therefore, it is necessary to answer several questions urgently, including, but not limited to, whether the infection of COVID-19 can be completely eliminated by adopting group immunization and what the relationship is between the threshold of group immunization and the virus reproduction constant *R*_0_ [1,2].

Predictive mathematical models play a crucial role in understanding the course of the epidemic and in designing strategies to quickly contain spreading infectious diseases in the face of the lack of any specific antivirals or effective vaccines [3,4,5,6]. For COVID-19 mathematical modeling, many attempts have been made to predict the epidemic trend based on the pioneering model developed by Kermack and McKendrick [7] in 1927, which is the original classic SIR model (namely susceptible (*S*), infected (*I*), and removed (*R*) classes). Most of these COVID-19 modeling studies achieved a good fitting result when overlined onto the statistical data. The models mainly extended the original SIR model by adding new compartments to get a more accurate simulation of the actual scenario. For example, Chen et al. [8] extended the transmission network into four compartments: bats, hosts, reservoir, and people. Kankan et al. [9] extended the SEIR (susceptible–exposed–infectious–removed) model by incorporating three additional compartments: asymptomatic (*A*), isolated infected (*Iq*), and quarantined susceptible (*Sq*). Subhas et al. [10] studied the epidemic in India with intervention strategies by using a mathematical model that consists of six subpopulations, susceptible *S*(*t*), exposed *E*(*t*), asymptomatic *A*(*t*), symptomatic or clinically ill *I*(*t*), hospitalized or isolated *H*(*t*), and recovered *R*(*t*) individuals, in a total population of *N*(*t*) = *S*(*t*) + *E*(*t*) + *A*(*t*) + *I*(*t*) + *H*(*t*) + *R*(*t*) individuals. Piu et al. [11] studied the COVID-19 transmission dynamics in India using a mathematical model with four compartments: the SAIU model, with SAIU standing for susceptible or uninfected (*S*) → asymptomatic (*A*) → reported symptomatic infectious (*I*) → unreported symptomatic infectious (*U*).

Through the model fitting process, some critical parameters can be estimated and then the virus basic reproduction number *R*_0_ can be derived based on these parameters using the concept of the next-generation matrix [12,13,14]. The *R*_0_ value could vary significantly in different regions. For example, *R*_0_ in different states in India varied in the range from 1.12 to 2.47 in the epidemic dynamics modeled by Kankan et al. [9]. *R*_0_ would also be different even for the exact same location, given a different period to fit the model. For instance, Wu et al. [15] studied an SEIR model to investigate the dynamics of COVID-19 infections based on the data from Wuhan, China, from 31 December 2019 to 28 January 2020 and calculated that the basic reproduction number was approximately 2.68. In a fractional-order model studied by Khan and Atangana [16] that simulates the dynamics of the COVID-19 outbreak in Wuhan, China, the authors computed the basic reproduction *R*_0_ as 2.4829 based on the data from 21 January to 28 January. 

The analysis of viral dynamics using mathematical models has helped gain an understanding of viral infections such as tuberculosis, dengue, and zika virus [17,18,19]. However, the majority of the classic ODE mathematical models, if not all, cannot predict the epidemic trend well, although most of them have an excellent fitting result. Here are some potential reasons for this. Firstly, suppose we fit the models by using the early surging section of the first epidemic wave. In that case, the parameter estimation process always returns a minimal *S*_0_ [9,10,11,12,13,14,15,16,20,21,22,23], which is unrealistic since we know *S*_0_ stands for the initial susceptible people in a particular population and it should be at least in the same order of magnitude as the entire population. The majority of the human population is susceptible to COVID-19 without vaccination, as revealed by experiments and cohort studies [24,25]. Secondly, in an opposite way, if we treat *S*_0_ as a constant approximately equal to the group population, using transmission parameters obtained by data fitting, we will find the epidemic growth rate is too sharp and the virus reproduction number *R* declines slowly. The epidemic growth rate in the early period is almost exponential. This prediction also contradicts the actual situation. Thirdly, while most of the traditional ordinary differential equation (ODE) models would be able to predict the infection’s turning point, few of them would be able to predict when and how the second wave or the third wave will begin. The configuration of ODE models with fixed parameters allows them to produce only one round of the epidemic. From our point of view, a crucial reason behind this drawback is the ignorance of the population’s geographic distribution. Without considering the spatial distribution characteristics of the population, it is difficult to accurately estimate the development of epidemic situations by using the traditional SIR model. ODE models with a fixed transmission coefficient face the challenge of providing more accurate and reliable prediction results. With the development of the COVID-19 epidemic, people gradually realized that the transmission coefficient is a varied term. To reproduce and fit the multiple-wave pattern of the epidemic trend, researchers are more inclined to adopt a revised compartment model. Most of the model revisions are concentrated on defining a time-dependent transmission coefficient. The attempts can achieve good fitting results, especially when handling the fluctuated epidemic situation [26,27,28,29,30,31]. Nevertheless, there are two major limitations of these approaches. Firstly, they lack physical background, especially to the critical problem of why the transmission coefficient β varies through time. However, without the derivation of the physical background, these equations are less likely to be ubiquitous and transformative to other cases. Secondly, adding more parameters typically returns better fitting results, especially on making some parameters time dependent. This may cause the issue of overfitting and damage the prediction capacity. Some ODE models have even integrated artificial intelligence approaches, such as the neural network, to further define the varied transmission coefficient [32,33,34], but it is still hard for these models to give a reliable prediction about when and how the next epidemic wave would occur. In particular, the driving forces at different epidemic stages are different. For instance, the second and third waves in the United States were mainly contributed by geographic diffusion. However, the fourth and fifth waves are mainly contributed by the vanishing immunity against reinfection (more details will be provided the Section 3).

The rapid development of computation power enables agent-based approaches for modeling complex systems with highly interacting individuals [35,36]. The influential modeling property of the agent-based model enables its wide application, such as in the optimization of supply chains [37], in the interpretation of corruption in ancient civilizations [38], and in modeling the dynamics of the immune system [39]. The agent-based (also called the individual-based) approach represents a new paradigm to model the spread of infectious disease and incorporate population heterogeneity and spatial information. In particular, agent-based models can make a more accurate and reliable prediction in conditions where it is required to forecast the development of the epidemic at a more fine-context level. Therefore, many agent-based strategies have been proposed to forecast the infection possibility of each element and the overall behaviors of the epidemic. For the study of COVID-19, Hoertel et al. proposed a stochastic agent-based model to simulate the early epidemic in France [40]. Hinch et al. built an agent-based framework named “OpenABM-Covid19” to study the non-pharmaceutical interventions against COVID-19 in the UK [41]. Cuevas proposed an agent-based model with position movement to evaluate the transmission risk of COVID-19 [42]. Under the agent-based methodology, several interesting basic global patterns have been proposed to simulate complex phenomena, such as diffusion, concentration and insolating, fire spreading, and segregation [43,44]. These behavioral patterns have been analyzed in terms of the simple rules that provoke them. The traditional agent-based model assumes that the agents can move freely within the environment. While this assumption can emulate the contact dynamics between agents, it has several critical disadvantages. First, the binary decision, which is represented as being infected or not, cannot accurately predict the epidemic trend, especially using a small-scale system. The simulation will return a stochastic result under the same initial condition per run. Second, the physical movement will add to the computational cost. Meanwhile, it does not obey the actual population interaction principles. To be specific, humans tend to interact with their neighbors around their living community. However, many agent-based models adopt a constraint-free movement, which will lead to significant position fluctuations after a certain period. Third, most of these models assume a life-long immunity to COVID-19 infection. Therefore, they will treat the recovered agent as not susceptible to infection. This assumption has been verified to be highly unreliable since tremendous breakthrough infection has occurred in COVID-19. 

To address the above limitations of traditional agent-based models, a continuous Markov-chain agent-based model that simulates the epidemic development was developed in this work. Applications of Markov chains can be found in many fields, from statistical physics to financial time series [45,46,47]. Here, we applied the Markov-chain approach to the prediction of the COVID-19 epidemic. The proposed model is distinct from the traditional agent-based model in three ways. Firstly, a continuous number range from 0 to 1 is used to represent the infection possibility so that the solution is unique and more accurate, even for a small-scale system. Secondly, the position of each element is assumed to be fixed to reduce the computational load and make it applicable for a more extensive system. Meanwhile, this assumption follows the actual patterns of human population movement, according to which most individuals are settlers instead of pastoralists. Finally, the proposed comprehensive Markov-chain model does not assume a life-long immunity existing toward COVID-19 infection. This is consistent with the experiments and cohort studies that indicate that the protection brought by infection or vaccination fades over time. Therefore, the recovered patient who has different susceptibility to reinfection is treated based on the time interval between the two infections. While the details of the proposed approach are given in Section 2, the three key components are summarized as follows: (1) a synthetic population that emulates the natural population’s demographic distribution, (2) a social contact network among the agents in the model where the contact frequency can be simplified to be negatively correlated with their distances, and (3) transmission rules, which could translate the edge weights in the interaction network into an overall infection probability toward each agent at specific time points. With these key components, the proposed model can take the information of population contact into account to simulate the spreading dynamics more accurately. Besides, this approach can integrate many features into the model, such as the virus mutation factor, population age distribution, public prevention, and control measures, to further generate a more reliable prediction. This model can predict the epidemic development in actual cases and provide valuable information for the development of the epidemic and the epidemiological tracking of infection cases. It can finally help explain the first questions: whether the infection in COVID-19 can be completely eliminated by adopting group immunization and what the relationship is between the threshold of group immunization and the virus basic reproduction constant *R*_0_ [1,2].

## 2. Methods

### 2.1. Derivation of the Markov-Chain Model for COVID-19 Infection

#### 2.1.1. Derivation of a Simplified Markov-Chain Model without Considering COVID-19 Mutation 

The continuous Markov-chain model of infection occurrence proposed in this work is briefly described below. It is assumed that there are N individuals in a population and there are different contact probabilities among those individuals. The infection probability is positively correlated with contact probability. For the simplified model, the relation constant is 1, which means the infection probability equals contact probability. The contact probabilities with themselves are zero. In this way, a matrix with N column × N rows is established, which has the following characteristics: (1)$$M_{interaction}\left(i,i\right)=0$$
(2)$$M_{interaction}\left(i,j\right)=M_{interaction}\left(j,i\right)$$
where $M_{interaction}\left(i,j\right)$ stands for the interaction possibility between individual *i* and individual *j*.

An accurate contact matrix can be obtained by tracking the individual contact probability in a natural population group. For example, each person’s mobile phone can be recorded to obtain the population contact matrix within a particular time phase. The contact matrix is temporal and dynamic, which means it changes over time. However, it is difficult to obtain such accurate data at present. Therefore, the contact frequency is determined according to the relative distance between individuals, as shown in Equation (3).
(3)$$M_{interaction}\left(i,j\right)=min\left(c_{1},\frac{c_{2}}{distance{\left(i,j\right)}^{n}}\right)$$
where *c*_1_ is the maximal contact possibility between agent *i* and agent *j*. A detailed explanation of *c*_1_ can be found in the model description section below, and *c*_2_ is a constant used to quantify the interaction possibility between two agents given a spatial distribution. The contact probability between two different individuals, the second term in the parenthesis on the right, equals the constant *c*_2_ divided by the *n*-th power of the distance between them. *c*_2_ and *n* can be adjusted in order to generate a reasonable *R*_0_ value, and if we use a larger *c*_2_ value and a smaller *n* value, we obtain a bigger *R*_0_ value correspondingly. In particular, the values of *c*_1_, *c*_2_, and *n* are preliminarily determined according to the initial reproduction constant *R*_0_ of the virus. More details will be provided in Section 2.2. The approach for calculating the basic reproduction number *R*_0_ is given below. 

The virus basic reproduction number *R*_0_ is the expected number of secondary cases produced, in a completely susceptible population, by a typical infective individual [12,13]. This dimensionless basic reproduction number is an essential indicator of the virus transmission ability. In a more general way, *R*_0_ can be stated as the number of new infections created by a specific infective population at a disease-free equilibrium and *R*_0_ represents the original transmission potential. As infections spread, the virus reproduction number *R* will decline while *R*_0_ is a fixed number. Therefore, the basic reproduction number *R*_0_ is not the same as the reproduction number of virus *R*. The basic reproduction number *R*_0_ can be computed using the next-generation matrix concept (see [12,13,14] for details). Since ordinary differential equations are not used in the proposed model, a new way is used to represent *R*_0_, that is, *R*_0_ is initially estimated by the contact frequency of people multiplied by the transmission coefficient of the virus in Equation (4):(4)$$R_{0}=\frac{1}{N}\sum_{i}^{N}\sum_{j}^{N}M_{interaction}\left(i,j\right)$$

The matrix $P_{infection}\left(i,t\right)$, which contains *N* rows for individuals of the studied group and *M* columns for the total number of generations, is defined to represent the probability of infection of individual *i* in the *t*-th generation of infection. The number *t* represents the current virus generation or the current time. It can be estimated through the following equations: (5)$$P1=P_{susceptibility}\left(k\right)=1-\sum_{i=1}^{t-1}P_{infection}\left(k,i\right)$$
(6)$$P2=\overset{N}{\underset{i=1}{\sum }}P_{infection}\left(i,t-1\right)\times P_{susceptibility}\left(k\right)\times M_{interaction}\left(i,k\right)$$
(7)$$P_{infection}\left(k,t\right)=\min\left(P1,P2\right)$$
where $P_{infection}\left(i,t-1\right)$ represents the probability of infection of individual *i* in the previous generation and $P_{susceptibility}\left(k\right)$ represents the susceptibility probability of individual *k* in the *t*-th infection generation, which is between 0 and 1. 

#### 2.1.2. Derivation of a Comprehensive Markov-Chain Model with Terms for Virus Mutation and Age Impact

To incorporate the impact of virus mutation on COVID-19 infection, a mutation term is added to Equation (5) for the susceptibility probability, as shown below.
(8)$$P_{susceptibility}\left(k\right)=1-\sum_{i=1}^{t-1}P_{infection}\left(k,i\right)\times M_{mutation}\left(k,i\right)$$
where $M_{mutation}\left(k,i\right)$ represents the attenuation effect caused by virus mutation and antibody attenuation and it is defined by Equation (9). If there is no antibody attenuation and virus mutation effect, then $M_{mutation}\left(k,i\right)$ = 1.
(9)$$M_{mutation}\left(k,i\right)={\left(1-r_{antibody\_fading}\right)}^{t-i}\times {\left(1-r_{mutation}\right)}^{t-i}$$
where $r_{antibody\_fading}$ represents the attenuation constant of the antibody with time (i.e., the number of infected generations). Since it represents the attenuation constant of a single generation, it is a small number. Similarly, the mutation rate $r_{mutation}$ represents the variation constant of a virus with time (i.e., the number of infected generations). Since it represents the variation constant during a single generation, it is also a small number. Although the values of these two constants are small, the iteration effect of several generations can cause a significant decrease in $M_{mutation}\left(k,i\right)$. For instance, if an individual gets infected at the (*t* − 1)-th time point, it will provide a strong protection against reinfection at the *t*-th time point because of the mutation term of ${\left(1-r_{antibody\_fading}\right)}^{1}\times {\left(1-r_{mutation}\right)}^{1}$, which is close to 1. However, this protection effect will vanish with time, as $M_{mutation}\left(k,i\right)$ will decrease with *k* increasing. Therefore, it will lead to a further increase in $P_{susceptibility}\left(k\right)$ according to Equation (8). Equation (8), together with Equation (9), is used to describe a vanishing immunity against reinfection, which is a key feature of the proposed model. 

The two constants $r_{antibody\_fading}$ and $r_{mutation}$ are estimated according to the following procedure: According to literature and news reports [48], the average infection cycle in COVID-19 is 7 days. The vaccine protection caused by the Indian mutant *B.1.617.2* is about 1 − 88% = 0.12, and the Indian mutant occurs around the 50th infection cycle. The decline in vaccine protection caused by the British mutant *B.1.1.7* is about 1 − 93% = 0.07, and the occurrence time of the British mutant is about the 30th infection cycle, so it is preliminarily inferred that $r_{mutation}$ = 0.002. According to the statistical data of reinfection after infection, for people under 65 years old, the average protection rate of preventing the second infection after infection within 50 infection cycles is 80%. We speculate that the protection rate after 50 infection cycles is much lower than 80%, calculated to be 70% [49]. Based on the mutation constants of viruses, we can preliminarily infer the antibody attenuation constant = 0.005.

In the proposed Markov-chain model, the relationship between age and infection probability is incorporated. A more accurate mathematical model should also take the influence of the population’s immune variation into consideration. Our model mainly considers the influence of age-related immunity vibration on infection risk. According to the statistical results of infection distribution at different ages, the relationship between infection probability and age is further derived as Equation (10).
(10)$$f\left(age\left(k\right)\right)={\left(1-\frac{1}{1+e^{\frac{age\left(k\right)}{25}}}\right)}^{2}$$
where f(age(k)) is a correction factor describing the susceptibility of the population of a certain age. It will increase as the age increases. The term age(k) indicates the age of the *k*-th individual in the population.

In addition, the dose effects on infection are considered in the model. According to our research, the occurrence of infection is related to the initial number of virus invasions. Therefore, for people with low infection probability, their contagious potentials are much smaller than those with high infection probability. The infected person does not necessarily have symptoms or even a positive diagnosis in the nucleic acid test. Therefore, a correction term $P_{transmission}\left(k,t\right)$ is added to quantify the relationship between the infection rate and the development of an individual into an infectious individual.
(11)$$P_{transmission}\left(k,t\right)=P_{infection}\left(k,t\right)\times 0.1^{\left(\frac{1-P_{infection}\left(k,t\right)}{5}\right)}$$
where $P_{transmission}\left(k,t\right)$ represents the transmission capacity of individual *k* at the *t*-th generation of infection. Equation (11) represents a nonlinear relationship between the infection possibility and its spreading capacity. For example, $P_{transmission}\left(k,t\right)$ becomes 1 if one individual has 100% infection possibility at time point *t*. It represents that the further spreading capacity of the virus would be 1. However, when an individual has a 50% chance of getting infected at time point *t*, the further transmission capacity will decline to 39.7% rather than 50% according to Equation (11).

After the aforementioned factors are considered, the final Markov-chain model turns out to be:(12)$$P1=P_{susceptibility}\left(k\right)=1-\sum_{i=1}^{t-1}P_{infection}\left(k,i\right)\times M_{mutation}\left(k,i\right)$$
(13)$$P2=f\left(age\left(k\right)\right)\times \overset{N}{\underset{i=1}{\sum }}P_{transmission}\left(i,t-1\right)\times P_{susceptibility}\left(k\right)\times M_{interaction}\left(i,k\right)$$
(14)$$P_{infection}\left(k,t\right)=\min\left(P1,P2\right)$$

Equation (12) differs from Equation (6) in that two correction terms are added, i.e., $P_{transmission}\left(i,t-1\right)$ and f(age(k)). Besides that, the comprehensive Markov-chain model returns a different value of $P_{susceptibility}\left(k\right)$ because of the addition of a vanishing immunity term described in Equations (8) and (9). 

Accordingly, the virus reproduction coefficient *R*_0_ becomes:(15)$$R_{0}=\frac{1}{N}\sum_{i}^{N}\sum_{j}^{N}M_{interaction}\left(i,j\right)\times f\left(age\left(i\right)\right)\times P_{transmission}\left(i\right)$$

### 2.2. Model Framework

The methodological section above briefly defined the function terms used in the model, and the equations are further used to describe the model framework here. The proposed model quantifies the possibility of a person being infected using a continuous probability rather than simply using two states (infected and not infected). Thus, for a population containing individuals {*a*_1_*, a*_2_ to *a_n_*}, the probability of each individual having the infection is a continuous number from 0 to 1. This is the rationale for defining the proposed model as a continuous Markov-chain model with the following steps:

Step 1, define and quantify the agent contact probability: We consider the probability that an individual *i* is infected at moment *t* − 1 as $P_{infection}\left(i,t-1\right)$ and the probability that this person infects individual *j* during an infection cycle as $M_{interaction}\left(i,j\right).$ Our model considers each individual to be location fixed. Fixing the location of each individual has the following advantage: in case we do not have access to the true frequency of contact in the population, we can establish the probability of transmission as a function of the distance between them to roughly calculate the probability of transmission between them. The disadvantage of using a dynamic model is that, while increasing the computational effort, moving individuals without restrictions will result in significant shifts in the locations of individuals in the population over time, which does not conform to the population movements in the actual scenarios. For most people, they have position alternations centered on the place of residence, so a static model of location is a better reflection of population contact.

The diagram of agent-contact probability is displayed in Figure 1. Figure 1A represents the contagion model with a distance constraint. For the continuous Markov-chain model with distance constraints, the element in the interaction matrix $M_{interaction}\left(i,j\right)$ is determined by Equation (3) if the distance between two individuals (i.e., indexed by *i* and *j*) is less than the threshold. Otherwise, a zero value is set to $M_{interaction}\left(i,j\right)$. The constant *c*_1_, a constant representing the maximal infection possibility from agent *i* to agent *j*, is a constant chosen from 0 to 1. It is set to 0.8 in the simulation based on the preliminary data for COVID-19. One rationale for this is that if two agents are too close to each other, such as ① and ②, the infection probability might exceed 1 if the term *c*_1_ is not incorporated in the model. This is unrealistic. Different combinations of *c*_1_*, c*_2_, and *n* would lead to a different *R*_0_ value. Instead, arbitrarily assigning values to these parameters, we have to guarantee that these parameters will generate a reasonable *R*_0_ value based on Equation (4). Figure 1B shows the contagion without the distance constraint, from which it can be seen that the probability of the virus transmitting to individual *j* from individual *i* is determined by $M_{interaction}\left(i,j\right)$ in Equation (3).

Step 2, quantify the contribution of agent *i* toward agent *j* in infection: Since the probability that agent *i* is infected at moment *t* − *1* is $P_{infection}\left(i,t-1\right)$ and the probability that agent *i* infects agent j during an infection cycle is $M_{interaction}\left(i,j\right),$ the value of the contribution of agent *i* to the infection of agent *j* is defined by Equation (13), i.e., $P_{infection}\left(i,t-1\right)\times P_{susceptibility}\left(j\right)\times M_{interaction}\left(i,j\right),$ where $P_{susceptibility}\left(j\right)$ is the coefficient of susceptibility of *j* to infection at moment *t*. This coefficient is illustrated in Table 1. Table 1 shows the calculation of the susceptibility coefficient $P_{susceptibility}\left(j\right)$ in a simplified Markov-chain model. As a simple example, suppose the calculated probability of infection of agent *j* in the previous *t* − 1 time periods is $P_{infection}\left(i,t-1\right)$, the corresponding susceptibility coefficient in the next time period is $1-\sum_{i=1}^{t-1}P_{infection}\left(j,i\right),$ as shown in Equation (5). This is based on the assumption of the lifetime immunity model, which indicates that the probability of a secondary infection for an infected individual after recovery is 0. To relax this constraint, Table 2 shows a schematic representation of the calculation of $P_{susceptibility}\left(j\right)$ for a more comprehensive Markov-chain model with more factors (e.g., mutation) considered, as shown in Equation (12). As a simple example, suppose the probability of infection calculated for agent *j* at the first *t* − 1 time periods is $P_{infection}\left(j,t-1\right)$. Then, the susceptibility factor of this person at the next time period is $1-\sum_{i=1}^{t-1}P_{infection}\left(j,i\right)*M_{\mathrm{mutation}}\left(j,i\right)$. This assumption is for the non-lifelong immunity model, which means that once an individual has been infected, the probability of a second infection after recovery is no longer 0. The susceptibility coefficient tends to increase over time. The susceptibility factor $P_{susceptibility}\left(j\right)$ is a continuous number from 0 to 1, and $M_{\mathrm{mutation}}\left(j,i\right)$ is expressed in Equation (9).

The difference between a simplified Markov-chain model (Equations (5)–(7)) and a comprehensive Markov-chain model (Equations (12)–(14)) is that the latter one considers other factors, such as the age (i.e., the term f(age[k])) and the viral dose (i.e., the term $P_{transmission}\left(k,t\right))$. As shown in Equation (10), the probability of infection and the probability of exposure in the comprehensive model show a certain exponential relationship and the probability of infection is not equivalent to the probability of exposure.

Step 3, calculate the probability of infection of individual *j* at time *t*: On the basis of the first two steps, an overall calculation of the probability of infection of individual *j* at time *t* is calculated in the last step. For the simplified model, this probability is quantified by Equation (7), while Equation (14) indicates the infection probability when the effect from mutation, age, and viral dose on infection is considered.

## 3. Results

### 3.1. An Illustration of the Simplified Markov-Chain Model 

The detailed description of the Markov-chain model is explained in Section 2. Here, we use a simple but concrete case to further illustrate our approach. Firstly, we study a simplified Markov-chain model without considering complicated factors. This model has the following assumptions: (1) the individual immunity to certain infectious diseases is homogeneous, and there is no individual variation, (2) there is neither virus mutation nor an antibody attenuation effect with time, (3) all infections will have the same transmission potential (i.e., if an individual is infected, they will produce antibodies and at the same time, they are contagious), and (4) individuals will recover after an infection cycle without death, that is, the overall population size will not change. 

Taking this model as an example, three individuals A, B, and C are listed in Table 3 for their infection possibility in the first few virus transmission cycles to be studied. According to the contact matrix shown in Table 3, the initial virus reproduction coefficient *R*_0_ = 1/3 × (0 + 0.8 + 0.5 + 0.8 + 0 + 0.6 + 0.5 + 0.6) = 1.267, according to Equation (4). If A gets sick firstly, the calculated infection probability of A, B, and C are shown in Table 4.

Our Markov-chain model is continuous. Each individual’s infection in a specific period is treated as a probabilistic problem rather than a simple infected or uninfected state, which would be represented as a Boolean number. The number of infected patients in a population at a particular time point is the sum of the infection probabilities of each individual. This probability could better reflect the actual epidemic dynamic when the population size scales up to a certain level.

### 3.2. The Prediction Capacity of the Markov-Chain Model Is Significantly Better Than That of the SIR Model

We expanded this model to 10,000 people. We randomly assigned the coordinates of these 10,000 people to the square zone with X-direction (0–250) and Y-direction (0–250). Using Equation (3), we calculated the contact matrix of the population. When *c*_1_, *c*_2_, and *n* were set to 0.8, 5, and 4, respectively, the initial virus reproduction number *R*_0_ was equal to 2 from Equation (4). The prediction of the epidemic curve was compared for three different model trails, and the results are shown in Figure 2.

It can be seen from Figure 2 that the early rising trend of the infection curve predicted by the SIR model is steep, while that of the Markov-chain model, considering population contact, is relatively mild. Our Markov-chain models are separated into two types. One is the constraint model, considering population contact distance. The contact probability of all people is inversely proportional to the fourth power of the distance between them. It is represented as the red curve in Figure 2. The other is a model with a population diffusion limitation, that is, within a specific distance range, the contact frequency of all people is inversely proportional to the fourth power of the distance between them. Besides that, the contact frequency of individuals beyond this distance threshold is set to be 0. This model is displayed as a blue dotted line in Figure 2.

The individual contact frequency of the actual scenario may be somewhere in between these two theoretical models. It can be seen from Figure 2 that the epidemic curve generated by the Markov-chain model with a population diffusion constraint rises slowly. The epidemic growth of the Markov-chain model without a population diffusion constraint is significantly milder than that of the SIR infection model. Compared with the SIR model, the infection curve predicted by the Markov-chain model better matches the actual infection statistical data. The simulation results generated by the SIR model may deviate significantly from the actual situation. One important reason for this deviation is that this model does not comprehensively consider epidemic development time and space factors, especially the influence of the population contact matrix brought by geographic population distribution on the overall infection curve. ODE models are inclined to forecast a significantly steep epidemic rising trend. However, forced data fitting often results in a tiny number of susceptible people, which is unrealistic compared with the actual situation, as stated in the introduction. The main reason for this phenomenon is that the traditional ODE model presumes that infected people have infinite flow or diffusion ability. After that, when only a small number of people are infected in a vast population, the change in the virus reproduction number *R* can be almost ignored, so the early epidemic prediction is often exponential. However, this is not the case in the actual situation. Even without any means of prevention and control, the epidemic’s growth may not follow the exponential trend, mainly due to diffusion’s spatial effect. To be more specific, the infected individual will prioritize causing infection to nearby people instead of equally transmitting the infection to people in random areas. This transmission mode will lead to a significant decrease in *R*-value even when only a small number of people are infected. This spatial effect of this epidemic-spreading dynamic can be well reflected by our Markov-chain model. As displayed in Figure 2, the infection curve predicted by the Markov-chain model can more accurately reflect the complicated epidemic fluctuation when multiple features, such as spatial population distributions, are considered. At the same time, our model can predict and track the hotspots during epidemic development, which is shown in Appendix A. It can effectively simulate the dynamic process of the epidemic situation at different geographic and time scales.

Generally speaking, one of the biggest problems with the compartment, or SIR, model is that it assumes a homogeneous system, which is the basis for using fixed parameters. Take the transmission coefficient β as an example. In reality, the spatial structure of the population causes the mobility of this system to be deficient, so the spatial structure causes the transmission coefficient to undergo a rapid decrement. This rapid decline starts much earlier before a significant decline in the number of susceptible people *S*. This sub-exponential growth is mainly contributed by a contact ratio decrement caused by a specific population spatial distribution. As a simple illustration, we assume a virus with *R*_0_ = 3 starts to spread in a large population. In this case, we assume one person can only come in contact with three neighboring people. The first person will infect three more people after the first infection cycle. However, the newly infected person will definitely infect fewer than two people because at least one of the neighbors has been infected. Therefore, the virus reproduction number will show a significant decline much earlier before a significant decline in the of overall susceptible population *S*. 

The SIR model and derivatives are the framework of choice to capture population-level processes. The basic SIR model, similar to many other epidemiological models, begins with an assumption that individuals form a single large population and that they all mix randomly with one another. This assumption leads to early exponential growth dynamics in the absence of control interventions and susceptible depletion and greatly simplifies mathematical analysis (note, though, that other assumptions and models can also result in exponential growth). The SIR model is often not a realistic representation of the human behavior driving an epidemic, however. Even in huge populations, individuals do not mix randomly with one another—they have more interactions with family members, friends, and coworkers than with people they do not know. This issue becomes especially important when considering the spread of infectious diseases across a geographic space, because geographic separation inherently results in nonrandom interactions, with more frequent contact between individuals who are located near each other than between those who are further apart. It is important to realize, however, that there are many other dimensions besides geographic space that lead to nonrandom interactions among individuals. For example, populations can be structured into age, ethnic, religious, kin, or risk groups. These dimensions are, however, aspects of some sort of space (e.g., behavioral, demographic, or social space), and they can almost always be modeled in a similar fashion to geographic space. However, it is difficult to divide the overall system into small systems that guarantee homogeneity among each small system. As we mentioned above, even if we divide the overall system into relatively small communities, such as different age groups or groups in different geographic spaces, we still cannot guarantee homogeneity among those sub-compartments because contacts happened at the unit level of the individual. If we divide the overall system in an extreme way, then it turns out to be an individual-based approach, which is the same as we used. This Markov-chain model is individual based. The infection possibility of each individual at a specific time unit is calculated based on their infection susceptibility and the infection possibilities of their neighboring agents. The infection susceptibility is calculated based on their infection possibility at historical time stages. This model can capture the infection possibility of each individual at discrete times. The overall epidemic development can be reflected as the summation of infection possibilities of all individuals. Since each individual is spatially located, this Markov-chain model can reflect the spatial distribution of epidemic hotspots at different times, as further described in Appendix A. 

### 3.3. There Is No Simple Derivation Relationship between the Virus Basic Reproduction Number R_0_ and the Final Herd Immunity Threshold

As shown in Figure 3, the threshold of group immunity predicted by different *R*_0_ values is significantly different based on different methods. The threshold of group immunity predicted by the SIR model is the highest, while the value of group immunity predicted by our Markov-chain model is also significantly higher than the value determined by *R*_0_ directly. For COVID-19, assuming its *R*_0_ is equal to 3, the threshold of herd immunity predicted by the simplified Markov-chain model is above 95%, significantly different from the 66.6% (derived from 1 − 1/*R*_0_) presumed by using the *R*_0_ value. The correct prediction of the group immunity threshold plays a vital role in guiding public policies such as vaccination coverage. The herd immunity threshold predicted based on the simplified Markov-chain model ignores many features, such as virus variation and individual immunity differences, so the predicted group immunity threshold is not necessarily accurate. However, our simplified Markov-chain model can reflect a problem. The simple method of inferring the group immunity threshold based on the *R*_0_ value may not be completely accurate and reliable. Although there is a significant positive correlation between the virus reproduction coefficient *R*_0_ and the group immunity threshold, the presumed relationship represented in the equation, i.e., threshold = 1 − 1/*R*_0_, is not satisfied. This presumed equation significantly underestimates the actual threshold of herd immunity in a natural infection event.

We further studied the influence of vaccination rate on the final number of infections. We assumed that the vaccination could be achieved instantaneously. The effect of vaccination on our model was equivalent to indirectly reducing population density. For example, assuming the vaccine is 100% effective, a 70% vaccination rate is equivalent to 3000 people randomly distributed in the original area instead of 10,000 people. The relationship between the vaccination rate and the final number of infected people is shown in Figure 4.

It can be seen from Figure 4 that for a virus with a basic reproduction number *R*_0_ = 3, on the premise of 100% vaccine effectiveness, there is a negative linear correlation between the vaccination rate and the basic reproduction number *R*_0_ after vaccination. The yellow curve represents the infection ratio of uninoculated people predicted by the Markov-chain model. The results indicate that 90% pre-vaccination coverage will cause a 1.2% infection probability in the remaining 10% population, 80% vaccination coverage will cause a 7.2% infection probability in the remaining 20%, 70% vaccination coverage will cause a 22.8% infection probability in the remaining 30%, and 60% vaccination coverage will cause a 40.9% infection probability in the remaining 40%. Since there is no available method in guiding the public to decide what vaccination coverage percentage is the best, the Markov-chain model provides a mathematical solution to this optimization problem.

In the method part, we compare the definitions and calculations of *R*_0_ between previous models and the models developed in this work. The reason why people tend to compare *R*_0_ with 1 originated from the pioneering work of Diekmann et al. [12], who pointed out that under an initial infection of 0, the value of *R*_0_ less than 1 would make the whole system in a disease-free equilibrium state. This background is the origin of a misunderstanding. *R*_0_ < 1 determines that an infected population creates less than 1 new infected population during its infective period and the infection can die out. On the contrary, *R*_0_ > 1 determines that each infected population creates, on average, more than 1 new infection and the disease can spread over the population. Therefore, in our public policymaking, policymakers linked the vaccination rate with *R*_0_. There is nothing wrong with the previous mathematic deduction, but people confuse the *R*_0_ value with the *R*-value. Once infection occurs, the virus reproduction number *R* will change constantly. The number of infected people and disseminators in the population increases and the virus reproduction number R is no longer equal to *R*_0_. Therefore, the concept of herd immunity should be interpreted as follows: for viruses with a reproduction number *R*_0_, assuming that the vaccine is 100% effective, an infection can be prevented given (1 − 1/*R*_0_) pre-vaccination population. However, it cannot halt infection when massive infections have already begun. Group immunity may not be explained in this way. In the case of natural infection, (1 − 1/*R*_0_) people will eventually become infected. For the case of natural infection, both the ordinary differential equation model and our Markov-chain model will predict an overall infection rate much higher than (1 − 1/*R*_0_). Then, it is reasonable to explain the phenomena that occurred in Iran, Brazil, Britain, and Israel. For example, the antibody-positive rate in Manaus had reached 76% in October 2020, but there was still a large-scale outbreak in December 2020 [50]. In some areas of Iran, where natural immunization was adopted, and in August 2020, in some provinces with a difficult epidemic situation, such as Rasht City, the positivity rate of the antibody had exceeded the threshold of herd immunity [51]. Meanwhile, countless reports indicate that massive infection could occur even at a group immunity level higher than 1 − 1/*R*_0_. For instance, a serum prevalence study in Britain [52] indicated that more than 93% of adults had antibodies against COVID-19 in late July 2021. However, mass infections were still happening in Britain. Israel [53] also noticed that 70% of vaccination coverage was not protective against COVID-19 infection. Figure 3 can calculate the proportion of people who caused the final infection under different initial vaccination proportions. More importantly, once a particular scale of infection occurs, the *R*-value can be quickly reduced by vaccination, but even if the *R*-value is reduced to a value below 1, it will be difficult for us to eliminate the virus in a short time. The higher the vaccination rate, the faster the virus elimination rate. We can no longer stick to the value of 1 − 1/*R*_0_, which does not reflect the threshold of group immunity. To sum up, the ratio of 1 − 1/*R*_0_ is the threshold of population immunity in the absence of any infection but not the threshold of population immunity to prevent infection. However, all those calculations assume a lifelong immunity for certain viruses and that herd immunity could be achieved. For COVID-19 infection, the situation is much more complicated, as shown in Section 3.4.

### 3.4. Simulation of the COVID-19 Epidemic Using a Complicated Markov-Chain Model

For realistic epidemic modeling, we often need to consider more variables, such as the influence of virus mutation and antibody attenuation effects, regional population distribution effects, and population age structures’ impact on the epidemic development. At the same time, another essential aspect that we must consider is the dose effect on infection probability, that is, the relationship between the infectivity and the initial amount of invading virus. A notable phenomenon in COVID-19 infection is the emergence of a large number of asymptomatic patients. Moreover, an interesting investigation feedback is that the proportion of serum prevalence is much higher than the reported number of infected people [54]. Experiments have confirmed that the severity of patients’ symptoms is positively correlated with viral load in vivo [55,56]. Our model shows that different infection degrees will possess different transmission potentials. The definition of “infection” in our model is based on the existence of antibodies. The individual with a small chance of being infected would have a smaller transmission potential. Therefore, a correction function (Equation (11)) was added to transform its infection possibility into the transmission potential after being infected. We simulated the epidemic dynamics of a specific region with a population distributed in four different cities. Most people in different cities have no contact opportunity with each other, except for a few of them. These few people become the links that connect the interactions in different regions. In the simulation, the following parameters were used: the mutation constant of the virus as 0.002, the attenuation constant of antibody as 0.005, the relationship between infection occurrence and age as in Equation (10), and the correction relationship between infectivity and contact probability as in Equation (11). Other specific parameters are provided in the Appendix A. The simulation results are shown in Figure 5.

From Figure 5A, we can see that although the public prevention policies could significantly affect the development of the epidemic situation, the geographic population distribution is an essential factor or even a dominant factor in driving the trend of the epidemic. Under a relatively stable public prevention strength, the spatial distribution of the population will lead to wave-like epidemic fluctuations and display multiple peak points. This trend was fully reflected in countries and regions suffering under the COVID-19 epidemic. Therefore, when we forecast epidemic development, we need to consider the spatial and geographical factors. A short-term decline does not necessarily indicate an overall decline in the epidemic situation but maybe a signal of the epidemic spreading from one region to another. From Figure 5B, we can see the spatial migration of infection hotspots more clearly. For example, at time point B, the infected people mainly concentrated in city 1. However, at time point C, the infection hotspot moved to city 2 and the epidemic situation in city 1 subsided to a certain extent. Another critical application of Figure 5A is to evaluate the impact of virus mutation and antibody decay on the epidemic development. The function of virus mutation and antibody decay we used is a simplified function that lacks sufficient data support, but it can roughly mimic the actual situation to a certain extent. Unlike the simplified model, modeling results predicted by the developed comprehensive Markov-chain model indicated that we might never reach herd immunity. We have to be prepared to coexist with COVID-19 for a long time because there is a possibility theoretically that the virus may not be completely eliminated by natural immunization or vaccination. Due to COVID-19’s natural attributes, for instance, high mutation rate and the existence of the antibody fading effect, which means antibodies produced by the human body fade away over time, the future epidemic may not have hotspots. However, it will be randomly distributed worldwide with a relatively small probability and COVID-19 will become a wildly disseminated disease. In our model, as shown in Figure 5G, at time point G, when a complete herd immunization cycle has been realized, the epidemic may have a reoccurrence in city 1. At that time, the epidemic situation is characterized by a small probability of a lack of concentrated hotspots and a wild distribution. As our whole human society, the complexity of its population, and its spatial distribution far exceed the scale of our model, this would provide a more favorable breeding ground for the evolution of viruses, so the probability of reinfection will significantly increase. The epidemic recurrence is already on its way, even after high vaccination coverage. This had already been confirmed by the third epidemic wave in Britain, starting from June 2021 [57]. According to epidemic data, a similar situation is still happening in some high-vaccination countries, such as the U.S. and Israel, and many other highly vaccinated countries [57]. This phenomenon is consistent with our prediction. A more vivid example is the local epidemic caused by the delta variant started on 20 July 2021, in China. Unlike the early epidemic cases, the government found it hard to trace the infection [58]. This epidemic has already been disseminated into many other provinces. It will take more effort and longer time to control the infection compared with several former cases. This is because the delta variant has a more robust transmission capacity and because the social immunity structure has been dramatically reshaped due to the massive vaccination process. Since more than 50% of the Chinese have been vaccinated against COVID-19, the scenario would resemble point G displayed in Figure 5G. A wild-spread diffusion characterizes the latter stages of the epidemic.

At the same time, we can predict the infection proportion in the overall population after the first round of infection through parameter estimation. This infection proportion does not simply correlate with *R*_0_, and it is closely related to the population age structure and the population contact matrix. For the simulation described above, the calculated virus basic reproduction coefficient *R*_0_ is 2.1175, corresponding to the traditional group immunity threshold of 52.7%. However, the actual serum prevalence reached 66.2% after a natural herd immunization cycle (1–210 generations), among which 1.3% of infected people had a second infection. A similar phenomenon has been reported by a serum prevalence study in Iran, indicating that at epidemic hotspots, the antibody-positive rate has further exceeded the herd immunity threshold derived directly from *R*_0_. Specifically, Nazemipour and colleagues stated that 72·6% seroprevalence in Rasht City did not follow the presumed herd immunity threshold [51]. Using the Markov-chain model, we can also calculate the serum prevalence in different age groups. We also noticed another interesting phenomenon. As shown by the purple dotted line in Figure 5, the average age of infection does not engage a significant alternation, which means that the virus’s infectivity to different age groups will not change in the spatial diffusion process. The change in the average infection age during the epidemic may be caused by other factors, such as the change of exposure frequency caused by age factors or some intrinsic features at the virus genome level.

While interesting findings have been shown in the simulation results, the real-world data are yet to be integrated into the proposed models due to the following two reasons. Firstly, it is not easy to obtain the accurate population distribution for large areas such as a state or a country and the communication fluxes need to be integrated among different regions. Another efficient way in the future is to track individual contacts based on their mobiles. However, it is also impossible for us to access this kind of data in the short-term future. Secondly, the computational cost is linearly proportional to the square of the agent number. Therefore, it is not feasible now to predict the actual epidemic trend at a population scale over 10 million. Nevertheless, the actual large-size epidemic development can be also well reflected in our small-scale modeling attempt. In Figure 6, we illustrate how our model can be used to roughly predict and interpret the real-world epidemic development. Here, we adopt the United States as an example. Figure 6 represents the epidemic development in the U.S., with the data extracted from one epidemic surveillance data site [57]. In general, the U.S., epidemics can be roughly categorized into five waves. There is an important time point, which is marked in Figure 6. That is the time point that herd immunity threshold had been reached, when over 2/3 of the American people had been vaccinated or infected. This time point is equal to the time point between time points F and G in Figure 5, which is displayed as a bottom point in daily infection in Figure 5. The second wave and the third wave were partly caused by the vanishing immunity against new infection and were mainly contributed by the epidemic diffusion effects, as shown from Figure 5B–F. To be more specific, the first wave was mainly contributed by the infections on the East Coast; the second wave was mainly contributed by the infections in big cities in the middle and on the West Coast; the third epidemic wave was mainly contributed by a further spreading to the rest of the country, especially rural areas. One of the most important implications of our approach is that it can predict the recurrence of an epidemic wave even after the so-called herd immunity threshold. Since our model is based on a vanishing immunity function, we predict that there would be new epidemic waves one after another even after massive vaccination coverage. The fourth and the undergoing fifth epidemic waves in the U.S. are consistent with the model prediction. In Figure 5A, we illustrate that there would be new epidemic resurgence, as shown at time points G, H, and I. Furthermore, nowadays, the epidemic spatial distribution has a good match with our prediction. We predicted that although the overall infections might be worse than the previous few epidemic waves, there would be no epidemic hotspots, as shown from Figure 5G–I. The infections after herd immunity threshold are more equally distributed within the whole country and are almost proportional to the population densities of certain areas. 

## 4. Discussion

John von Neumann had a great saying: “With four parameters I can fit an elephant, and with five, I can make him wiggle his trunk.” He meant that one should not be impressed when a complex model fits a data set well. With enough parameters, a researcher can fit any data set. Compared with the traditional SIR method, the SIR method with increased parameters, including various susceptible–infected–recovered–death (SIRD) models, susceptible–exposed–infected–recovered–death (SEIRD) models, etc., can achieve good fitting results [8,9,10,11,20,21,22,23,59,60,61]. Nevertheless, these models all fall into the trap of pure mathematical fitting. Using multi-parameters, one can produce better fitting results. However, multiple parameters can also bring several critical problems: Firstly, the solution of parameters is not unique. Secondly, the lack of a robust physical mechanism does not have a good prediction effect, confirmed in many early studies of the COVID-19 epidemic. These classic models can rarely accurately predict the inflection turning point of the epidemic, let alone the repeated fluctuations of the epidemic. With the development of the compartment model with time-delay distribution, people gradually realized that the transmission coefficient is a time-varying term. Many research groups have done excellent research on recapturing the resurgence of epidemic. However, as we discussed in the introduction part, these attempts lack physical background. It is hard to judge which factors dominate the change in the transmission coefficient. Is the alteration in the transmission coefficient due to the lockdown, the population distribution, or the fluctuating human immunity against this virus? To answer this question and to give a better prediction about the future epidemic development, we decided to adopt an agent-based approach that can integrate more specific individual information into the entire system, including people’s age and their geographic distribution. 

Based on this idea, we established a Markov-chain model of virus infection for the first time. Our model can effectively consider the impact of the actual contact probability of the population on the epidemic development. The population contact probability matrix can be roughly calculated according to the spatial population distribution. Besides that, we can further integrate in vivo individual contact frequencies reflected in the actual situation into our model. Compared with SIR and other ordinary differential equation system models, the proposed Markov-chain model can integrate more information, such as the contact frequency of different individuals, closely related to spatial location and individual relationship.

Meanwhile, it can comprehensively consider the virus mutation effect, the antibody attenuation effect, the population age structure, and other factors. These advantages endow this model with better capacity for robust prediction, evidenced by the model’s prediction of the epidemic dynamics through time and the detection of the epidemic hotspot distribution at different times. If we could access accurate data for analysis, we could further refine the parameters in our Markov-chain model. We could also effectively forecast the spatial and temporal trends of epidemic situations and predict the herd immunity threshold using these parameters. We have to reiterate that the herd immunity threshold does not have a simple relationship with its *R*_0_. The actual herd immunity threshold might be significantly higher than the presumed one derived from *R*_0_. This finding might significantly influence the future public decision, which indicates that higher vaccination coverage needs to be reached to rapidly reduce infection cases. There is still space in our model for improvement. For example, the computational cost is proportional to the square of the population size. Although our model has a great potential to stimulate more realistic statistical data, we have not applied it to an in vivo scenario due to the availability of the data. The future research mainly includes improving the algorithm efficiency, integrating in vivo data to obtain more reliable parameters, and verifying the reliability of this method in the analysis of confirmed cases.

## 5. Conclusions

In this study, we developed a continuous Markov-chain model to simulate the dynamic behavior of the COVID-19 epidemic. This Markov-chain approach can be classified as one of the agent-based approaches in which individual contacts are considered. Therefore, the heterogeneity and the population spatial distribution can be considered. Using this model, we demonstrated that the herd immunity threshold (1 − 1/*R*_0_) could not provide robust protection against infection at the population level. The herd immunity threshold should be significantly larger than the previous estimation even of the life-long immunity assumption. Furthermore, taking the immunity waning effect, the complex Markov-chain model can predict the multi-wave trend of the COVID-19 epidemic both at the time and spatial levels. Although the virulence of SARS-CoV-2 might show a significant alteration in the future, it is challenging to eradicate SARS-CoV-2 in a short time without strong interventions. This Markov-chain model can be treated as a robust alternative method to the traditional compartment modeling approaches.

## Figures and Tables

**Figure 1 biology-11-00190-f001:**
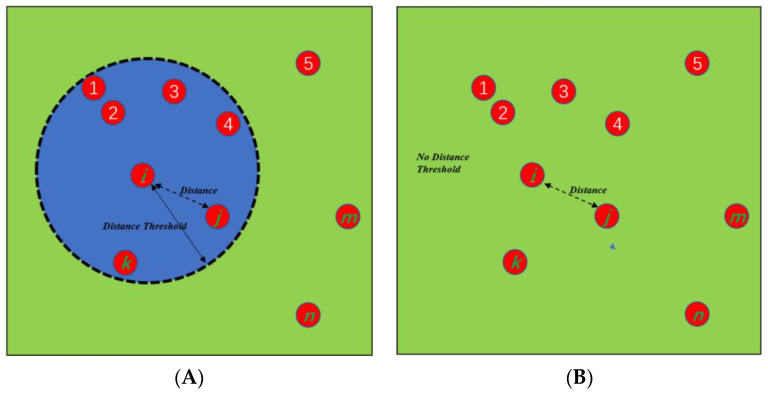
An illustration of a constraint-based Markov-chain model with a distance threshold (**A**) and an illustration of a constraint-free Markov-chain model (**B**).

**Figure 2 biology-11-00190-f002:**
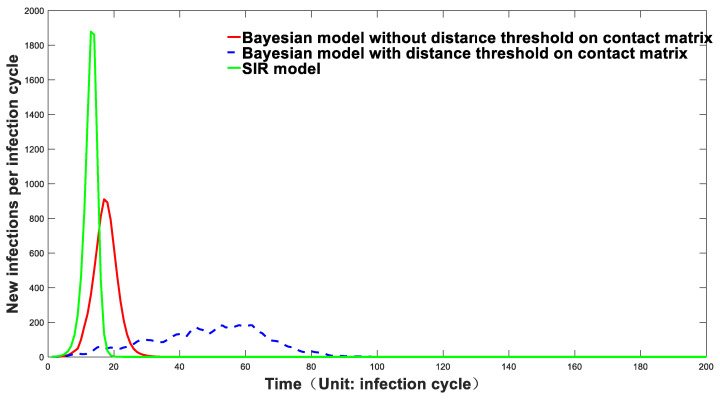
Epidemic trend predicted by three different models.

**Figure 3 biology-11-00190-f003:**
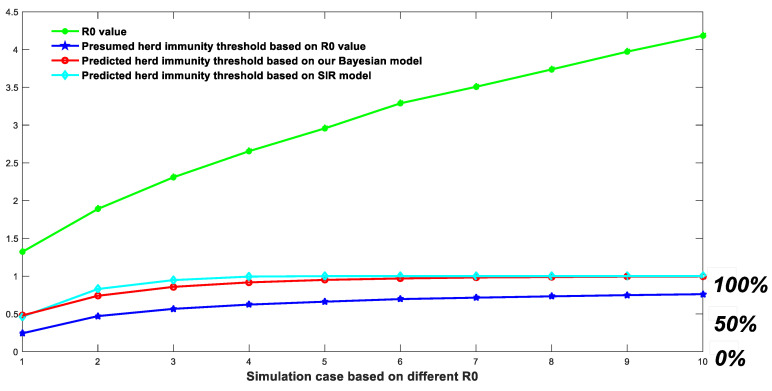
Herd immunity threshold predicted by three different approaches.

**Figure 4 biology-11-00190-f004:**
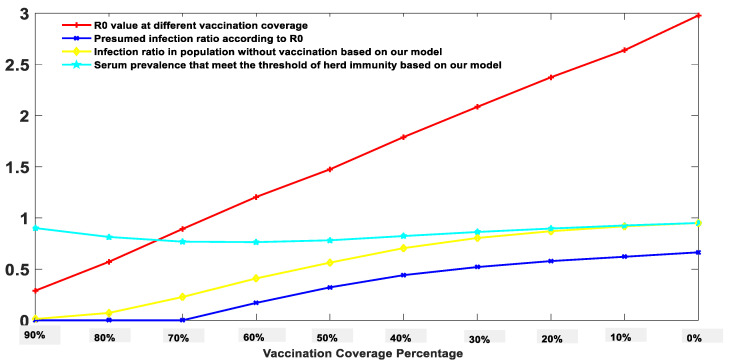
Predicted infection probability at different vaccination coverage percentages using a simplified Markov-chain model.

**Figure 5 biology-11-00190-f005:**
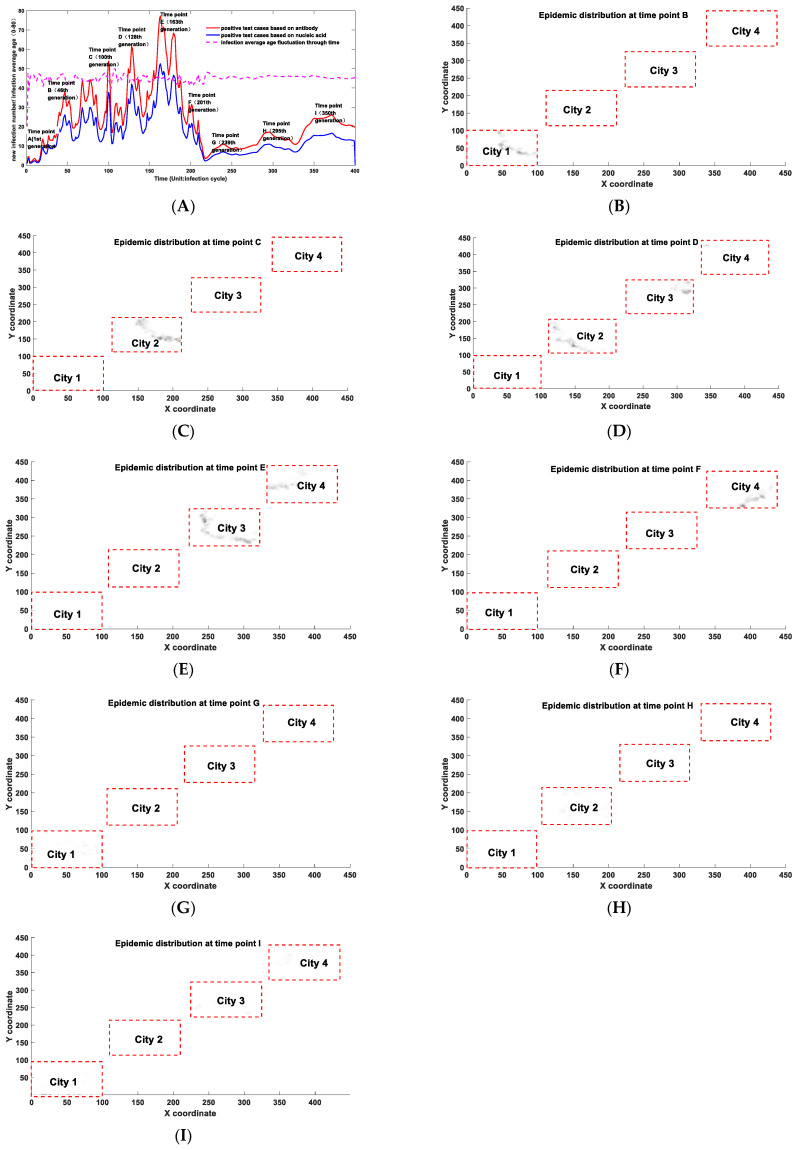
Epidemic trend predicted by the Markov-chain model with the consideration of multiple factors (**A**) and the predicted epidemic distribution at time point B (**B**), at time point C (**C**), at time point D (**D**), at time point E (**E**), at time point F (**F**), at time point G (**G**), at time point H (**H**), and at time point I (**I**).

**Figure 6 biology-11-00190-f006:**
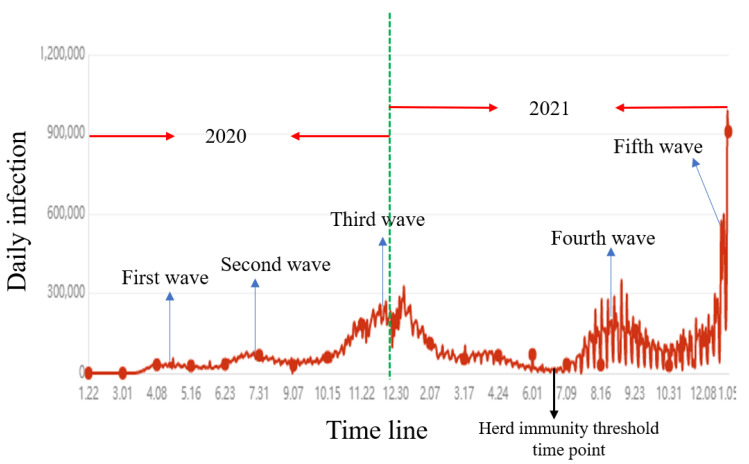
An implication of our prediction for the real-world scenario: an interpretation of the epidemic in the United States.

**Table 1 biology-11-00190-t001:** An illustration of definition of ${\mathrm{sus}}_{\mathrm{probability}\left(j\right)}$ of agent *j* in the simplified Markov-chain model.

Time Points	Infection Possibility	$P_{susceptibility}$
1	$P_{infection}\left(j,1\right)$	1
2	$P_{infection}\left(j,2\right)$	1 − $P_{infection}\left(j,1\right)$
3	$P_{infection}\left(j,3\right)$	1 − $P_{infection}\left(j,1\right)-P_{infection}\left(j,2\right)$
*k*	$P_{infection}\left(j,k\right)$	$1-\sum_{i=1}^{k-1}P_{infection}\left(j,i\right)$

**Table 2 biology-11-00190-t002:** An illustration of definition of ${\mathrm{sus}}_{\mathrm{probability}\left(j\right)}$ of agent *j* in the complex Markov-chain model.

Time Points	Infection Possibility	$P_{susceptibility}$
1	$P_{infection}\left(j,1\right)$	1
2	$P_{infection}\left(j,2\right)$	1 − $P_{infection}\left(j,1\right)\times M_{mutation}\left(j,1\right)$
3	$P_{infection}\left(j,3\right)$	1 − $P_{infection}\left(j,1\right)\times M_{mutation}\left(j,1\right)-P_{infection}\left(j,2\right)\times M_{mutation}\left(j,2\right)$
*k*	$P_{infection}\left(j,k\right)$	$1-\sum_{i=1}^{k-1}P_{infection}\left(j,i\right)\times M_{mutation}\left(j,i\right)$

**Table 3 biology-11-00190-t003:** Interaction frequency matrix among three individuals.

Interaction Matrix	A	B	C
A	0	0.8	0.5
B	0.8	0	0.6
C	0.5	0.6	0

**Table 4 biology-11-00190-t004:** Infection probability of three individuals at different time points.

	A	B	C
1st generation	1	0	0
2nd generation	0	0.8	0.5
3rd generation	0	0.06	0.24
4th generation	0	0.02016	0.00936
*N*th generation	0		

## Data Availability

The data presented in this study are available in the Appendix A.

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
