# Peer review of "A Continuous Markov-Chain Model for the Simulation of COVID-19 Epidemic Dynamics"

_biology, 2022, doi:10.3390/biology11020190_

Round 1
Reviewer 1 Report
In this manuscript, the author introduced a Bayesian model for predicting the spread of SARS-CoV-2. The introduction completely describes the traditional mathematical models applied to predict epidemic dynamics. Moreover, the authors pointed out the issues of these conventional models. Building on this, the author suggested that considering the spatial distribution characteristics of the population could improve the forecasting properties of epidemic dynamics modelling. Thus, they introduced a continuous Bayesian agent-based model that emulates the natural population’s demographic distribution, a social contact network, and transmission rules. The model is exhaustively explained, and the results are complete. My only remark is that the authors have applied their model to simulated data. Thus, it would be necessary to assess the model and its predictions on real-world data, not just ad hoc scenarios. The authors should add a paragraph in the results section, which illustrates the prediction of SARS-CoV-2 spread of their model, based on Covid-19 real-world data collected so far or at least explain how their simulations are related to real-world data and comment about this.
Author Response
The responses are attached.

Reviewer 2 Report
The paper offers probabilistic modeling of COVID-19 growth using the matrices of contacts/infectiousness.
Comments:
- The introduction should be improved. You need to include related papers on COVID-19 modeling with the help of differential equations (primarily, differential equations with distributed delay).
- lines 88-91. The criticism of ODE models is not evidenced. I do not agree that "... none of them could be able to predict when and how the
second wave or third wave begins". For example, delayed differential equations are able to predict epidemic outbreaks. - lines 94-95. What do you mean "without considering the spatial distribution characteristics of the population, it is difficult to accurately estimate the development of epidemic situations by using the traditional SIR model."? SIR including diffusion (i.e., partial differential equations) allows us to develop a spatial-temporal epidemic model.
- The quality of the mathematical expressions should be improved throughout the paper.
- The description of the method should be improved significantly. line 180. What is the meaning of the N*N matrix?
- line 191. What are c1 and c2?
- Expression (4) for R0. Does your notion of the basic reproduction number fit the traditional notion which is calculated with the help of the next-generation approach? It should be so. If no, use another notion.
- Figure 2 is unclear. What do you mean "infection cycle"? How did you compute it for the ODE models?
- Discussion should be improved.
- Describe experimental data properly.
- The presentation and the derivation of the main model should be revised significantly. All terms and parameters should be specified.
Author Response
The responses are attached.

Reviewer 3 Report
Тhe proposed manuscript is devoted to the results of the authors related to a new continuous Bayesian agent-based model that simulates COVID-19 epidemic development. In the proposed model the infection possibility is described by a number within the continuous interval 0 to 1. The position of each agent is supposed to be fixed. The model does not assume a life-long immunity against COVID-19 infection and takes into account its fading through time. Important features of the epidemic dynamics are included into the presented model. The properties of the model are studied and compared with other approaches.
The presentation of the main results is clear and comprehensive. From a formal point of view, all the contents seems to be correct. The results are valuable and worthy of being published taking into account their possible applications for prediction of the epidemic development and better understanding of its mechanisms and possible methods of counteraction.
Minor revisions are suggested to improve the quality of the exposition:
- p. 1, line 39: I suggest to write “whether the infection with/of COVID-19 can be completely eliminated” instead of “whether the infection in COVID-19 can be completely eliminated”
- p. 2, line 55: I suggest to write “by using a mathematical model, which consists of six subpopulations” instead of “by using a mathematical model consists of six subpopulations”
- p. 2, line 64: I suggest to write “next-generation matrix” or “next generation matrix” instead of “nextgeneration matrix”
- p. 2, line 74: I suggest to write “The analysis of viral dynamics” instead of “Analysis of viral dynamics”
- p. 2, line 76: I suggest to write “the majority of the mathematical models” instead of “the majority of our mathematical models”
- p. 2, line 80: The meaning of S_0 should be described
- p. 3, line 99: The expression “and thus affect the higher-level organization appears of the entire system” is not very clear and should be reformulated
- p. 3, line 136: I suggest to write “it does not obey the actual population interaction principles” instead of “it does not abbey the actual population interaction principles”
- p. 3, line 150: I suggest to write “properties, according to which” instead of “properties, which”
- p. 4, line 172: I suggest to write “Terms definition” instead of “Term definition”
- 4, line 182 and later: The equations should be written more carefully
Author Response
Responses are attached.

Reviewer 4 Report
In this paper, the authors propose a Bayesian model to describe COVID-19 propagation, considering contact probabilities, mutation probabilities, etc. The topic is of interest and the paper could be considered a valuable contribution. However, the current version of the manuscript cannot be considered for publication. The manuscript needs substantial improvement in terms of grammar and structure. The description of the mathematical model is poor, it is hard to follow the derivation and justification of the model (especially because of the style of the Equations, but not only). The titles of Sections 3.1, 3.2, etc. are unsuitable (too long).
Some more specific comments:
- I don’t see the definition of c1 and c2 in Equation (3). I understand that they are constants to be determined. Please specify better the meaning of these constants.
- It is necessary to explain better the content of Figure 5, Figure 5A, etc. Besides, I don’t think this kind of numbering of the figures is standard.
- The authors state that the prediction capability of their model is greater than that of the SIR model. However, I don’t see any specific analysis of model prediction capability.
- The authors do not provide details on model implementation. The authors insist on the Bayesian nature of their model, but I cannot find specific details on the Bayesian formulation of the model (likelihood, prior distributions, etc.).
Author Response
Responses are attached.

Round 2
Reviewer 2 Report
All my comments were addressed except the introduction section. I recommend adding to the related works:
- https://doi.org/10.1007/s11071-020-05863-5 describing the usage of distributed delays model for COVID basing on statistical data
- https://doi.org/10.1109/ACCESS.2021.3104519 describing two-strain model and parameter identification basing on Big Data technology
Author Response
We added those two references as you recommended.
Reviewer 4 Report
The authors have improved some aspects of their study, but I am still unsure about some specific relevant points (the Bayesian nature of their model and its prediction capability). Therefore, I cannot change my previous decision.
Author Response
The responses are attached.

Round 3
Reviewer 4 Report
Now the paper is more coherent with the methods presented. Therefore, I think that it could be accepted for publication.